# Is Fresh Produce in Tigray, Ethiopia a Potential Transmission Vehicle for *Cryptosporidium* and *Giardia*?

**DOI:** 10.3390/foods10091979

**Published:** 2021-08-25

**Authors:** Tsegabirhan Kifleyohannes, John James Debenham, Lucy J. Robertson

**Affiliations:** 1Institute of Paraclinical Sciences, Faculty of Veterinary Medicine, NMBU, 1430 Ås, Norway; john.debenham@nmbu.no (J.J.D.); lucy.robertson@nmbu.no (L.J.R.); 2College of Veterinary Medicine, Mekelle University, Mekelle 2084, Ethiopia

**Keywords:** contamination, fruit, protozoa, vegetables, foodborne

## Abstract

In rural Ethiopia, where people often share their homes with their livestock, infections of humans and animals with *Cryptosporidium* and *Giardia* are relatively common. One possible transmission route is consumption of contaminated fresh produce; this study investigated the occurrence of *Cryptosporidium* and *Giardia* in fresh produce in four districts of rural Tigray in Ethiopia. Fresh produce samples (*n* = 55) were analysed using standard laboratory procedures. Overall, 15% (8/55) of samples were found to be contaminated. Although contamination levels were mostly low, a few samples had high numbers of *Giardia* cysts (up to around 70 cysts per 30 g sample). Molecular analyses were largely unsuccessful, but *Giardia* Assemblage A was identified in one sample. Contamination with these parasites was identified in two of the four districts, but, although a similar pattern has already been described for water contamination, this may be at least partially explained by sampling bias. Nevertheless, we speculate that access to clean water sources may be an important factor for reducing the occurrence of these pathogens. Given the public health and veterinary burden associated with both parasites, the factors which are of importance for their circulation in the communities and environments deserve further investigation.

## 1. Introduction

Foodborne parasitic diseases are an important public health concern globally [1] resulting in significant morbidity and mortality among susceptible populations [2]; the disease burden is particularly high in low and middle-income countries (LMIC) [2]. In the developed world, such as European countries, although the disease burden is lower, foodborne parasites are nevertheless also considered important, because of the challenges in monitoring, prevention, and control [3]; issues which are also relevant in LMIC.

According to WHO estimates of disease burden, foodborne parasitic diseases, excluding enteric protozoa, caused an estimated 23.2 million cases and 45,927 deaths in 2010, resulting in an estimated 6.64 million Disability Adjusted Life Years (DALYs) [2]. The same study showed that an additional 67.2 million cases, 5560 deaths, and 492,000 DALYs were due to foodborne enteric protozoa [2].

Fresh produce can be contaminated by enteric protozoan parasites, including *Cryptosporidium* and *Giardia* [4]. Contamination of fresh produce by parasites can occur pre-harvesting [5] during cultivation, via irrigation with contaminated water or sewage or the use of animal or human faeces as fertilizer. Contamination could also occur post-harvesting, by being washed with contaminated water or from handling by infected food-handlers [4,6], or during transportation and storage [5].

The survival capabilities of *Cryptosporidium* oocysts and *Giardia* cysts in moist and refrigerated environments means that fresh produce is a suitable transmission vehicle [4]. *Cryptosporidium* oocysts are more robust than *Giardia* cysts, and oocysts can survive under less favourable storage conditions for more extended periods [7]. *Cryptosporidium* oocysts that have contaminated fresh produce during harvesting may be still infective at the marketplace where it is sold for human consumption. Although, *Giardia* cysts are more fragile and may not survive less-favourable environmental conditions [4], contamination of fresh produce, particularly post-harvesting, may still lead to foodborne transmission. The risk of infection by *Cryptosporidium* and *Giardia* from contaminated fruits like berries can be significantly decreased by simple washing before consumption [8]. However, due to the adherent nature of some parasites, washing does not completely remove all of them [7,8,9], and on some fresh produce rough surfaces, crevices or hairs may entrap parasites [10].

In developing countries, like Ethiopia, enteric protozoan infections, including *Cryptosporidium* and *Giardia,* are common [11]. Both parasites are well-known causes of gastro-intestinal illness worldwide [12]. Diarrhoea is the leading cause of mortality in children younger than 5 years in Ethiopia, accounting for more than 70,000 death annually, which 23% of all under-five deaths [13]. In addition, cryptosporidiosis is of particular concern in the immunocompromised [14], or those suffering from other health insults such as malnutrition [15]; given that both malnutrition and AIDS occur at higher rates in countries of Sub-Saharan Africa than in many other global regions this is of particular concern [16]. The proportion of this diarrhoea caused by zoonotic protozoan parasites is not well understood.

Investigations of parasitic contamination of vegetables and fruits collected from local markets have been conducted in various towns in Ethiopia [5,17,18,19,20,21,22]. All these studies used a washing procedure followed by microscopy (direct wet mount, iodine wet mount, and modified Ziehl-Neelsen staining) to identify parasitic contamination. In these studies, the proportion of fresh produce contaminated by *Cryptosporidium* oocysts has been reported to range from 4.7–12.8% and with *Giardia* cysts from 1.3–18.5% indicating relatively high occurrence of contamination [5,17,18,19,20,21,22]. One limitation of these studies was quantitative data on the level of contamination of the produce samples was not reported, nor was the recovery efficiencies of the methods used provided. Additionally, no attempt was made to speciate the identified parasites by molecular methods, which is necessary to determine their threat to public health. Thus, although several studies have indicated that contamination of fresh produce with *Cryptosporidium* and *Giardia* is relatively common in Ethiopia, the importance of this contamination to human health remains unknown.

In most of Ethiopia, fruits and vegetables are sold in open-air marketplaces and streets, and often this produce is eaten directly after purchase without washing (personal observation during sample collection). Fresh produce displayed for sale without washing has been reported to be more likely to be contaminated with parasites than washed fresh produce [5]. Fresh produce is usually carried to Ethiopian marketplaces by people or animals, or in animal-drawn carts; public transport or private vehicles are sometimes used. According to Alemu et al. 2020 [5], the fruits and vegetables transported to the marketplace by animals were more likely to be contaminated with parasites than fresh produce transported in motor vehicles [5].

As part of a larger One Health study on the epidemiology of cryptosporidiosis and giardiasis among humans and animals in rural areas of Tigray, Ethiopia, we investigated contamination of vegetables and fruits collected from farmers’ backyards, irrigated farmlands, and open-air local markets at four locations. In the study areas, family labour was used for all the tasks, from land preparation to harvesting and transporting of the products to the market, organic manure (faeces of the animals) used to improve production.

Our aim was to determine the occurrence of *Cryptosporidium* and *Giardia* in fresh produce at four locations in Tigray, Ethiopia, to assess the species and genotypes of these parasites in order to provide information on transmission pathways and epidemiology, and to compare these data with the occurrence of the parasites in water samples previously reported from the four sampling districts.

## 2. Materials and Methods

### 2.1. Study Areas

Between October 2018 and January 2019, samples of fresh produce (vegetables and fruits commonly eaten raw) were collected from local open-air markets, backyards of farmers, and irrigated farmlands from four selected districts of Tigray Region, Ethiopia, namely: Enderta (south-eastern zone of the region), Kilte Awulaelo (eastern zone), Hintalo Wejirat (south-eastern zone), and Raya Azebo (south zone).

Enderta (Location 1) is located at 13°14′ N and 39°40′ E with an altitude ranging from 1500 to 2300 metres above sea level (masl). The district covers a total area of 89,812 km^2^ of which 30,062 ha is cultivable land [23]. The total population of the district both in rural and urban areas is approximately 114,300 according to the 2015 population and housing census data [24]. Location 1 comprises of two agro-climatic zones. In the main zone, the mean annual maximum and minimum temperature is 24 °C and 11 °C, respectively with an average annual rainfall of 601 mm [25]. A minor portion in the eastern and western parts has an elevation between 500 to 1500 masl, with an average temperature above 20 °C. The area is characterized by erratic rainfall and frequent droughts. The long rainy season is between June and September and the short rainy season is from March-May [25].

Hintalo Wejirat (Location 2) is at latitudes between 12°55′ N and 13°20′ N and longitudes 39°20′ E and 39°55′ E with an elevation area ranging from 1400 to 2850 masl. The district covers an area of 193,309 ha with an approximate total population of 153,500 [24,26]. The annual mean temperature in the area is 18 °C. There are two rainy seasons in the district, the long rainy season from June-August and the short rainy season from March-April. Rainfall patterns can be erratic and in the north-west of the region average annual rainfall is up to 850 mm, decreasing to 300–400 mm in the east [26].

Kilte Awulaelo (Location 3) is located between 13°45′ N–14°00′ N and 39°30′ E–39°45′ E. The elevation ranges from 1980 to 2500 masl. The average daily air temperature ranges between 15 °C and 30 °C. The mean annual rainfall is 601 mm. The district covers an area of 101,758 hectares, of which 21,620 hectares are farmlands [27,28]. The total population is approximately 99,700 [24].

Raya Azebo (Location 4) is located at 12°39′ N latitude and 39°44′ E longitude. The altitude ranges from 930 to 2300 masl. The area has two rainy seasons, with light rains between February to April period and heavy rains between July–September. The mean annual rainfall is 724 mm, with mean daily maximum and minimum temperatures of 18 °C and 14 °C, respectively for the western highlands and 23 °C and 20 °C, respectively in the valleys [29]. The district covers an area of about 176,210 ha with a population of 135,870 [24].

The majority of households do not cultivate vegetables, rather relying on local markets to purchase fresh produce. In Tigray, for example, production of vegetables and fruits is reported in only 14% and 8% of households, respectively [30]. Historically, most cultivated land is used to grow various types of cereal crops, with only a small portion dedicated to vegetable production. This trend is changing, however, with an increase in vegetable production since 2014 [31]. According to Tigray agricultural marketing promotion agency, an estimated 90% of the overall vegetables grown in the region are sold in the local market and the remaining 10% are used at home for family consumption [31].

### 2.2. Sample Collection

Of the vegetables and fruits available for raw consumption in the regions studied, the most common ones are cabbage, lettuce, carrot, pepper, tomato, and guava. These are the fresh produce types included in our study. The samples were collected from the four districts which are subdivided into ‘tabias’ (villages). Twenty-four samples were analysed from Location 1 (4 tabias), five samples from Location 2 (1 tabia), twenty-one samples from Location 3 (4 tabias), and five samples from Location 4 (1 tabia) (Table 1). The number of samples collected from each district varied, being affected by the schedule of the local markets and relatively low vegetable production, such that fresh produce was not always available for sampling when we were visiting each location. The total numbers of samples analysed for each type of fresh produce were: cabbage (12), carrot (6), lettuce (13), pepper (9), tomato (9), and guava (6) (Table 1).

Study participants were interviewed regarding their sources of vegetables and fruits for consumption. Based on responses, samples of fresh produce were collected from the relevant sources. The participants reporting that they ate vegetables produced in their own backyards provided vegetable samples from this location. Some participants reported that their vegetables and fruits sources came from their own irrigated fields, and these samples were obtained from their fields. In all cases, the collected samples were put in a plastic bag, coded with date, district, and site of collection, and transported to the parasitology laboratory, College of Veterinary Medicine, Mekelle, Ethiopia.

### 2.3. Preparation of the Sample for Analysis

The samples were refrigerated at 4 °C for a maximum of 24 h at the parasitology laboratory and processed on the following day. For leafy vegetables (lettuce and cabbage) and peppers, 30 g were weighed into stomacher bags (Seward BA6041/STRfilter bag), 200 mL of 1 M glycine buffer was added to immerse the sample, and the samples were processed by stomacher/paddle blender for one minute. For carrots, tomatoes, and guava one or two were selected on the basis of their size and put into filtered stomacher bags and washed for 5 min by hand in 200 mL of 1 M glycine buffer [4].

The eluate was transferred into five 50 mL centrifuge tubes, the bag washed with distilled water, and the wash water transferred to the tubes. The tubes were transported to Ayder Referral Hospital, Department of Microbiology, Mekelle on the same day, where they were centrifuged for 10 min at 1550 relative centrifugal force. Following aspiration of the supernatant, the pellets were vortexed and combined in a single tube per sample and refrigerated at 4 °C before being transported to Norway for further analysis.

### 2.4. Immunomagnetic Separation (IMS)

IMS was performed using Dynabeads GC-combo kit for isolation of *Cryptosporidium* oocysts and *Giardia* cysts. Although 16 samples from Location 1 district were processed according to the ISO 18 744 [32] protocol, all other samples were processed following a reduced-cost protocol that is based on this standard, but uses fewer beads (20 µL of each bead type), and modified buffers (as well as the buffers provided with the kit, PBS-Tween and SurModics StabliZyme AP buffer); the efficiency of the modified method is comparable to that of the ISO Method (30–50%) but is considerably cheaper due to the reduction in use of reagents [33].

### 2.5. Detection of Cryptosporidium Oocysts and Giardia Cysts Using Immunofluorescent Antibody Staining (IFAT)

Single-well slides of air-dried sample concentrates were fixed with methanol and stained with fluorescein isothiocyanate (FITC)-conjugated monoclonal antibodies (mABs) against *Cryptosporidium* oocyst walls and *Giardia* cyst walls (Aqua-glo™, Waterborne™ Inc., New Orleans, LA, USA) and 4′,6′ diamidino-2-phenylindole (DAPI) was used to stain the DNA in the nuclei of these organisms. Samples were mounted with M101 No-Fade Mounting Medium then each slide was covered by a glass coverslip and viewed immediately.

A Leica DCMB fluorescence microscope equipped with Nomarski differential interference contrast (DIC) optics was used for examination of the slides. A blue filter block (480 nm-excitation, 520 nm-emission) was used to visualize FITC conjugated mABS labelled cysts and oocysts and a UV filter block (350 nm excitation, 450 nm emission) was used to visualize the presence or absence of DAPI-stained sporozoite nuclei. All observations were at 200× or 400× magnification.

The entire well of the stained slides was examined for the presence or absence of *Cryptosporidium* oocysts and *Giardia* cysts; preliminary identification was based on reactivity with the monoclonal antibody and appropriate shape and size. Ovoid or spherical objects with brilliant apple-green fluorescence of the appropriate size and shape were examined under the UV filter to determine the presence of nuclei, both to support identification (if present) and for assessing the sample suitability for further investigation by molecular methods. The number of *Cryptosporidium* oocysts and *Giardia* cysts observed per sample was recorded.

### 2.6. Recovery Efficiency of the Method

The recovery efficiencies of both the methods used were estimated using spiked samples in which known numbers of flow cytometry-sorted oocysts and cysts (AccuSpike™-IR; Waterborne Inc., New Orleans, LA, USA and EasySeed™, TCS Biosciences Ltd., Botolph Claydon, UK) were spiked onto 30 g lettuce and dried at room temperature for 2 h before analysis as described. Recovery efficiencies were estimated to be approximately 30% for *Cryptosporidium* and 55% for *Giardia*, regardless of whether the ISO 18744 [32] protocol or reduced-cost method was used [33]. This is within the range considered acceptable using the ISO or US EPA methods for analysis of water samples.

### 2.7. DNA Extraction

DNA extraction was conducted on positive samples using DNeasy PowerSoil Kit (Qiagen, Oslo, Norway) protocol, with some modifications. The fresh produce sample post IMS (250 µL) and 60 µL of the lysis solution (solution C1) were added to the PowerBead Tubes and vortexed together to mix. This was then subjected to bead beating to release the DNA by breaking the oo(cyst) walls using a FastPrep-24 5G (MP Biomedicals) in two cycles of 4 metre/s for 60 s with a 45 s pause between the cycles. In the end, the DNA was eluted in 40 µL of the elution solution (solution C6) and stored at −20 °C. A recent study by Temesgen et al., 2021 [8] reported that the bead-beating approach is effective for obtaining the DNA of *Cryptosporidium* and *Giardia* from artificially contaminated berries.

### 2.8. Polymerase Chain Reaction and Sequencing

For *Giardia* investigation, PCR targeting the glutamate dehydrogenase gene, beta-giardin gene, and SSU gene were conducted according to published protocols [34,35]. For *Cryptosporidium*, PCR targeting the SSU-rRNA gene was conducted [35]. Details of primers and cycling conditions are listed in the Appendix A. Negative and positive controls were included in each PCR run.

Reactions were carried out in a total volume of 25 µL that included 2 µL of template DNA, 0.4 µM of each primer and 12.5 µL of DreamTaq PCR Master Mix (2X) (Thermo Fisher Scientific, Oslo, Norway). Bovine serum albumin (BSA) 0.2 µL of (20 mg/mL) was used in the reaction targeting the beta giardin and GDH genes.

The PCR products were examined following separation on a 2% agarose gel, stained with SYBR Safe DNA gel stain and visualized under UV illumination. A ready-to-use DNA ladder (Thermo Scientific, Oslo, Norway) of 100 bp was used for fragment size determination.

Purification of the positive products was carried out using ExoSAP-IT PCR product clean-up reagent (Thermofisher Scientific, Oslo, Norway) and sent to a commercial company (EUROFINS GENOMICS, Ebersberg bei München, Germany) for sequencing in both directions. Sequences were checked using Geneious prime software and compared with sequences in GenBank using NCBI BLAST.

### 2.9. Statistics and Data Handling

Results were collected in a database in Excel. Data analysis was made using simple descriptive statistics and frequency using STATA version 15. Fisher’s exact test was used to determine associations with sample location and detection of parasites and also between sources of samples and detection of parasites.

## 3. Results

### 3.1. Occurrence of Cryptosporidium and Giardia on Vegetables and Fruits

Out of 55 fresh produce samples processed, 8 (15%) were found to be contaminated with *Cryptosporidium* oocysts and/or *Giardia* cysts. DAPI staining was not observed in any of the oocysts or cysts, indicating that the oocysts or cysts had ruptured and there was no longer nuclear material within the oocysts or cysts. Of these 8 samples, 2 were contaminated with *Cryptosporidium* only, 5 were contaminated with *Giardia* only, and 1 was contaminated with both *Giardia* and *Cryptosporidium*. All 8 positive samples were from Location 1 and Location 3 districts, which were also the districts from which most samples were collected.

The contamination rate of different types of fresh produce is depicted in Figure 1. The contamination rate for cabbage was 3/12 (25%), lettuce 2/13 (15%), guava 2/6 (33%) and pepper 1/9 (11%). Contamination with these parasites was not detected on either carrot or tomato samples.

The number of contaminating oo(cysts) per sample of vegetable and fruits are described in Table 2. The degree of contamination was low for both parasites, with the exception of one cabbage sample where 71 *Giardia* cysts were counted and one lettuce sample where 28 *Giardia* cysts were counted; all other positive samples had fewer than 20 oo(cysts).

Of the positive samples, 1 (*Giardia* positive) was obtained from a backyard (of 5 total backyard samples), 3 from farmers’ fields (1 *Cryptosporidium* positive, 1 *Giardia* positive, and 1 with both parasites of 14 field samples), and 4 from markets (2 *Giardia* positive and 2 *Cryptosporidium* positive of 36 market samples). No source was more or less likely to be contaminated with these parasites than another.

### 3.2. Genotyping of Cryptosporidium and Giardia

DNA was isolated from the three samples with the highest number of *Giardia* cysts (71, 28, and 11 cysts), in the latter of which *Cryptosporidium* oocysts were also detected, and also from the two other *Cryptosporidium* positive samples.

PCR resulted in amplification at the *Giardia* SSU gene from the sample containing 28 cysts (lettuce sample from Location 3, Kilte Awulaelo). DNA sequencing of this PCR product was successful, and revealed it to be Assemblage A, with 99% similarity to sequences at GenBank Accession numbers MH047247.1 and MH047246.1. All other PCR attempts for the other *Giardia* and *Cryptosporidium* positive samples were unsuccessful despite multiple attempts at several genes.

### 3.3. Comparison between Occurrence of Positive Samples in Fresh Produce and Water in the Four Districts

In our previous article [36], contamination of drinking water with *Cryptosporidium* and *Giardia* appeared to occur frequently in Locations 1 (Enderta) and 3 (Kilte Awulaelo). These are also the two locations where contamination was found in fresh produce (Table 3). In addition, the fresh produce sample with 71 *Giardia* cysts came from Location 1, and a high count of *Giardia* cysts (22 cysts in 10 L) was also observed in one water sample from this location [36].

## 4. Discussion

This study investigated the extent of contamination of fresh produce with the protozoan parasites *Cryptosporidium* and *Giardia* in four districts of rural Tigray, northern Ethiopia. The most important finding from this study is that contamination of fresh produce with these parasites occurs relatively frequently in this area, with 8 out of 55 samples (15%) positive for *Cryptosporidium* and/or *Giardia*. Contamination was only found in two districts, Location 1 and Location 3, with a district-specific occurrence of 4/24 (17%) and 4/21 (19%), respectively. However, it should be noted that considerably more samples were analysed from these districts (45 samples in total from Locations 1 and 3, and 10 samples in total from Locations 2 and 4). This sampling bias means that we cannot reach a conclusion on whether the contamination risk was higher in Locations 1 and 3 than in Locations 2 and 4.

Other reports from Ethiopia on contamination of fresh produce have produced similar results regarding the proportion of contaminated fresh produce. For example, Alemu et al., (2020) [5] reported 4.9% and 10.2% occurrence of *Cryptosporidium* spp. and *Giardia* contamination, respectively, from vegetable and fruit samples collected from local markets in Bahirdar city, northwest Ethiopia. Similarly, Bekele et al., 2017 [20] reported a prevalence of *Cryptosporidium* spp. and *Giardia* of 4.7% and 10%, respectively, from vegetables and fruits collected from local markets in Arba Minch town, southern Ethiopia. Moreover, Endale et al., 2018 [37] reported 7.7% and 9.3% prevalence of *Cryptosporidium* spp. and *Giardia*, respectively, from vegetables and fruits collected from local markets in Dire Dawa, Eastern Ethiopia. Furthermore, Tefera et al., 2014 [22], reported contamination with *Cryptosporidium* spp. (12.8%) and *Giardia* (7.5%) from fruits and vegetables collected from selected local markets of Jimma town, Southwest Ethiopia. Nevertheless, the only similar study that we could find from Tigray reported a higher occurrence than we found with 7.3% and 18.5% occurrence of *Cryptosporidium* and *Giardia* on fresh produce from markets in the town of Aksum [19]. Discrepancies between our data and others from Ethiopia could be due to real differences in the occurrence of these parasites in different geographical locations but could also reflect differences in the laboratory analysis method. As IFAT is both more sensitive and more specific than traditional microscopy methods used in these other studies, both false-negative and false-positive results should be lower.

As previously noted, none of these other studies from Ethiopia quantified the extent of contamination. Although in our study the number of parasites detected per sample was usually low (mostly fewer than 5 (oo)cysts per 30 g sample), a few of the samples were quite heavily contaminated, with one sample harbouring over 70 *Giardia* cysts. Reports from other parts of the world also generally indicate low numbers of parasites among contaminated fresh produce. For example, a study from Norway reported 1–8 *Cryptosporidium* oocysts or *Giardia* cysts in 100 g samples of various types of fruits and vegetables [38]. More relevant to this current investigation is a study of vegetables from retail in India, where among positive samples of fresh produce obtained from street vendors, parasite (*Cryptosporidium* and *Giardia*) counts were usually below 10 per 30 g sample [4]. However, as in our study, some samples had very high counts [4], although, unlike in our study, this tended to be for *Cryptosporidium* oocysts rather than *Giardia* cysts. Another variable of interest is the type of fresh produce examined. In this study we analysed samples of vegetables commonly eaten raw (cabbage, carrot, lettuce, pepper, tomato) and guava. Although only a few guava samples were analysed, it was found to be the fresh produce most likely to be contaminated, with *Cryptosporidium* in 33% (2/6) and *Giardia* in 17% (1/6) of guava samples, followed by cabbage, lettuce, and pepper. Neither *Cryptosporidium* nor *Giardia* were detected on tomatoes and carrots. The reason for this may reflect factors associated with cultivation (carrots are below the surface of the soil and therefore less likely to be directly contaminated) or other factors, such as transport. As tomatoes are more likely to be damaged than the other fresh produces, they may be transported in containers and are therefore less likely to be contaminated during transport. Cabbage and lettuce may be more likely to be contaminated due to their high surface area, and their rough surfaces [5,7,20]. The higher contamination of guava is interesting, as they are on trees and thus unlikely to be contaminated in the field; thus, contamination during or post-harvest seems most likely.

None of the other studies from Ethiopia have investigated the species (*Cryptosporidium*) or genotype (both *Cryptosporidium* and *Giardia*) contaminating fresh produce. This information may provide clues on the source of contamination, and indicate the potential public health significance. Although we were only able to obtain data from one sample of *Giardia*, the finding of Assemblage A, which is known to infect humans and animals, opens for the possibility of multiple different transmission routes leading to contaminated produce, including zoonotic transmission.

Water analysed from all four study districts for these parasites also indicated water contamination at Locations 1 and 3 (Enderta and Kilte Awulaelo), but not at the other two locations [36]. However, as for the fresh produce samples, relatively few water samples were analysed from Locations 2 and 4. At Locations 1 and 3, there were more water sources available for sampling, whereas at Locations 2 and 4 several of the water sources were out of order (e.g., hand pumps) and at Location 4 there was a main common water source which limited sample collection. Although this sampling bias for both water and fresh produce could have influenced our results, it is also possible that these parasites are circulating more in the environment in some locations than others, and it would be of interest to examine this in greater detail. One possible explanation for the relative absence of environmental contamination at location 4 (Raya Azebo—Mechare tabia) is that the main water source was in-house tap water. Such a water supply is less likely to be contaminated by people and animals, and less likely to spread contamination to fresh produce or act as a transmission vehicle to those drinking it. Water supply has previously been shown to be a risk factor for diarrhoeal disease [39].

As with many studies, there are some important limitations in this study. The total number of fresh produce samples that we analysed was relatively low (55) compared to other studies, including those from Ethiopia where reported sample sizes were between 108 to 384 [5,17,18,19,20,21,22,37]. As our fresh produce samples were collected along with other samples, we were limited by resources. In addition, most of our study participants bought fresh produce from the market, which occurred only once per week, so opportunities for sample collection were restricted. In addition, we had challenges in our molecular analyses, and this reflects the relatively few positive samples, low concentrations of parasites, and a lack of nucleated parasites (as demonstrated by inclusion of DAPI) in those samples that were positive. Although DAPI staining was absent in the parasites detected on all positive vegetables, we speculated that the DNA may be in the suspension following elution of the samples and therefore conducted the molecular analyses. The nuclei could have been lost during transport, storage, and processing, particularly from *Giardia* cysts that are not as robust as *Cryptosporidium* oocysts [7]. Our molecular result from one of *Giardia* positive sample corroborates this suggestion, as DNA amplification was possible. It is not possible to determine whether the parasites contaminating the fresh produce were viable or infectious at the time of sample collection, and therefore presenting a public health risk to consumers. The lack of oo(cyst) nuclei at detection suggests that the parasites were non-infective. However, it is possible that damage to the oo(cyst)s occurred after sample collection, such as during transport, storage, or processing, and were indeed infective at the time of sampling.

In conclusion, our study found that contamination of fresh produce with *Cryptosporidium* oocysts and *Giardia* cysts was relatively common in Tigray region. Despite the fact that contamination was only detected in two of the four districts in the study, this could reflect sampling bias rather than a lack of contamination at Locations 2 and 4. Although contamination levels in positive samples were generally low, a few samples were more heavily contaminated with *Giardia* cysts. Similar results have already been reported regarding contamination of water supply, with the same two districts apparently having cleaner water. Although our data may suggest certain factors in specific locations may contribute to the circulation of these parasites within the environment, sampling bias also has a role. Given the public health and veterinary burden associated with both parasites, trying to pinpoint the factors of importance in reducing the circulation of these parasites in the communities and environments deserves further investigation.

## Figures and Tables

**Figure 1 foods-10-01979-f001:**
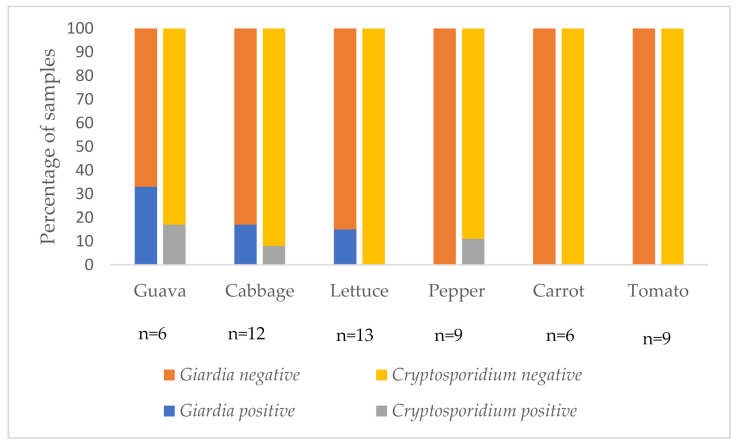
Proportion of *Giardia* and *Cryptosporidium* contaminated fresh produce samples.

**Table 1 foods-10-01979-t001:** Sources of vegetables and fruits from the four locations.

	Backyard	Fields	Market	Total
Enderta (Location 1)				
Cabbage	0	5	2	7
Carrot	0	0	2	2
Guava	1	0	2	3
Lettuce	0	2	3	5
Pepper	1	0	3	4
Tomato	0	0	3	3
**Total**	2	7	15	24
Hintalo Wejirat (Location 2)				
Cabbage	0	0	1	1
Carrot	0	0	1	1
Guava	0	0	0	0
Lettuce	0	0	1	1
Pepper	0	0	1	1
Tomato	0	0	1	1
**Total**	0	0	5	5
Kilte Awulaelo (Location 3)				
Cabbage	0	1	2	3
Carrot	0	0	2	2
Guava	0	2	1	3
Lettuce	2	2	2	6
Pepper	1	0	2	3
Tomato	0	2	2	4
**Total**	3	7	11	21
Raya Azebo (Location 4)				
Cabbage	0	0	1	1
Carrot	0	0	1	1
Guava	0	0	0	0
Lettuce	0	0	1	1
Pepper	0	0	1	1
Tomato	0	0	1	1
**Total**	0	0	5	5
**Grand total**	**5**	**14**	**36**	**55**

**Table 2 foods-10-01979-t002:** Intensity of contamination of fresh produce with *Cryptosporidium* oocysts and *Giardia* cysts.

Sampling Area	Type of Fresh Produce	Number of *Cryptosporidium* Oocysts Counted	Number of *Giardia* Cysts Counted
Location 1	Pepper	2	0
	Lettuce	0	3
	Cabbage	0	71
	Cabbage	0	3
Location 3	Lettuce	0	28
	Guava	0	1
	Guava	3	11
	Cabbage	2	0
Total positive samples		3	6

**Table 3 foods-10-01979-t003:** Occurrence of contamination of fresh produce with *Cryptosporidium* and *Giardia* oo(cysts) compared with water in the same locations.

	Proportion (%) of Samples Positive			
	Fresh Produce Samples		Water Samples	
	*Cryptosporidium*	*Giardia*	*Cryptosporidium*	*Giardia*
Location 1	4% (1/24)	13% (3/24)	7% (1/15)	40% (6/15)
Location 2	ND ^1^ (0/5)	ND (0/5)	ND (0/7)	ND (0/7)
Location 3	10% (2/21)	14% (3/21)	8% (1/13)	ND (0/13)
Location 4	ND (0/5)	ND (0/5)	ND (0/2)	ND (0/2)

^1^ ND-Not detected.

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
