# Peer review of "Is Fresh Produce in Tigray, Ethiopia a Potential Transmission Vehicle for Cryptosporidium and Giardia?"

_foods, 2021, doi:10.3390/foods10091979_

Round 1

Reviewer 1 Report

In this manuscript authors show a snapshot of the prevalence of two important food-borne transmitted parasites in the rural populations of Ethiopia. The topic is very interesting and some useful data about transmission and prevention could be informative for the Ethiopian health authorities. 

The paper in general is well organised, the method are adequate, and the manuscript well written and easy to follow. My only concern is that the number of samples collected is very very low to be consider a proper study (n= 55). Although the authors showed that the data obtained (% of prevalence) are in line with other publications, these publication used a >350 samples. I would suggest to unify all the location as one, and just stay with Backyard, fields and markets categories. 

Minor points: authors should make sure the references matches with the number given, for example: line 323 the Bekele et al is not the reference 13 is the 17.

Author Response

Please see the attachment. The line numbers are matching with the clean copy of the manuscript.

Reviewer 2 Report

The authors present a work on the occurence of Cryptosporidium and Giardia on fresh produce. Even though the subject is not novel, the fact that the research focuses on Ethiopia is new information. The results are clearly presented.

Please add title and unit in y axis (percentage?) in Figure 2 

Reviewer 3 Report

Dear Madam / Sir.

This article is about the occurrence of Cryptosporidium and Giardia in fresh products and water from the area of Tigray, Ethiopia. It is an interesting article since data concerning foodborne pathogens are of interest since in several cases foodborne parasites are underreported. The manuscript is generally well written, with appropriate structure. The methodology followed is adequate and described in detail. The only drawback of this work is the number of samples examined that is inconsistent and generally small, and does not allow accurate conclusions to be drawn in all the questions asked. However, this does not substantially reduce the value of the data reported.

The Introduction part is rather large. Approximately three paragraphs are sufficient for introducing the reader to the case.

P1 L15-16. n=8 can be added in here

P1 L20-22. This is a rather large sentence. Please simplify.

P1 L30-32. This is not something completely different from low and middle income countries. In addition, some data concerning disease burden in these countries can be added. Finally, no mention to the immunocompromised is made. To my knowledge, in low income countries the number of persons suffering from a disease or condition causing immune system deficiency is rather large. Since giardiasis and cryptosporidiasis are especially severe in immunocompromised persons, they can really be a problem in these countries.

P2 L57-62. Merge this paragraph to the one at the start (L27-32).

P2 L82-94. These paragraphs can be shortened or transferred to the Discussion part.

P4 Table 1. This table can be omitted. The information given, if dimmed necessary, can be added in the text.

P4 L186-189. Perhaps a short description of the modified protocol would be of interest.

P5 L215-216. A short description of the spiking procedure, including the cyst origin, is needed.

P7 L258-259. Do the authors mean that the DAPI staining was unsuccessful in recognizing the nuclei structure?

P7 Figure 1. The negative columns can be omitted. It is more confusing than helpful to the reader.

P7 L271. Replace “detailed” with “described”.

P8 Table 3. The table needs to be designed according to the journal’s recommendations. In addition, although it is perfectly acceptable and scientifically sound to add “ND”, it is less comprehensive to the reader. Just a suggestion though.

P9 L338-341. The authors use the world relative without giving a comparison with some standard values. Therefore, the word should be omitted or the values should be added. In the following paragraph, the cyst counts are also reported as low.

P9 L338-350. These two paragraphs should be merged. The second is a natural continuance of the first.

P10 L387-404. This paragraph, except from the part concerning DAPI (L396-404) should be omitted.

Author Response

Please see the attachment. The line numbers are matching with the clean copy.
